# Using the Prediction Model Risk of Bias Assessment Tool (PROBAST) to Evaluate Melanoma Prediction Studies

**DOI:** 10.3390/cancers14123033

**Published:** 2022-06-20

**Authors:** Isabelle Kaiser, Sonja Mathes, Annette B. Pfahlberg, Wolfgang Uter, Carola Berking, Markus V. Heppt, Theresa Steeb, Katharina Diehl, Olaf Gefeller

**Affiliations:** 1Department of Medical Informatics, Biometry and Epidemiology, Friedrich-Alexander-Universität Erlangen-Nürnberg, 91054 Erlangen, Germany; isabelle.kaiser@fau.de (I.K.); annette.pfahlberg@fau.de (A.B.P.); wolfgang.uter@fau.de (W.U.); katharina.diehl@fau.de (K.D.); 2Department of Dermatology and Allergy, Technische Universität München, 80802 München, Germany; sonja.mathes@mri.tum.de; 3Department of Dermatology, Universitätsklinikum Erlangen, Friedrich-Alexander-Universität Erlangen-Nürnberg (FAU), 91054 Erlangen, Germany; carola.berking@uk-erlangen.de (C.B.); markus.heppt@uk-erlangen.de (M.V.H.); theresa.steeb@uk-erlangen.de (T.S.); 4Comprehensive Cancer Center Erlangen-European Metropolitan Area of Nuremberg (CCC ER-EMN), 91054 Erlangen, Germany

**Keywords:** risk prediction, prediction models, risk of bias, PROBAST, melanoma

## Abstract

**Simple Summary:**

The rising incidence of cutaneous melanoma over recent decades, combined with a general interest in cancer risk prediction, has led to a high number of published melanoma risk prediction models. The aim of our work was to assess the validity of these models in order to discuss the current state of knowledge about how to predict incident cutaneous melanoma. To assess the risk of bias, we used a standardized procedure based on PROBAST (Prediction model Risk Of Bias ASsessment Tool). Only one of the 42 studies identified was rated as having a low risk of bias. However, it was encouraging to observe a recent reduction of problematic statistical methods used in the analyses. Nevertheless, the evidence base of high-quality studies that can be used to draw conclusions on the prediction of incident cutaneous melanoma is currently much weaker than the high number of studies on this topic would suggest.

**Abstract:**

Rising incidences of cutaneous melanoma have fueled the development of statistical models that predict individual melanoma risk. Our aim was to assess the validity of published prediction models for incident cutaneous melanoma using a standardized procedure based on PROBAST (Prediction model Risk Of Bias ASsessment Tool). We included studies that were identified by a recent systematic review and updated the literature search to ensure that our PROBAST rating included all relevant studies. Six reviewers assessed the risk of bias (ROB) for each study using the published “PROBAST Assessment Form” that consists of four domains and an overall ROB rating. We further examined a temporal effect regarding changes in overall and domain-specific ROB rating distributions. Altogether, 42 studies were assessed, of which the vast majority (*n* = 34; 81%) was rated as having high ROB. Only one study was judged as having low ROB. The main reasons for high ROB ratings were the use of hospital controls in case-control studies and the omission of any validation of prediction models. However, our temporal analysis results showed a significant reduction in the number of studies with high ROB for the domain “analysis”. Nevertheless, the evidence base of high-quality studies that can be used to draw conclusions on the prediction of incident cutaneous melanoma is currently much weaker than the high number of studies on this topic would suggest.

## 1. Introduction

Cutaneous melanoma is one of the most lethal forms of skin cancer that accounts for the majority of skin cancer deaths [1]. The incidence rates of melanoma have been growing dramatically over recent decades in most fair-skinned populations worldwide with annual increases of 3 to 7% [2,3,4]. The highest incidence rates by far are observed in Australia and New Zealand [5], although the incidence rates in these two countries are now stabilizing or even slightly declining following intensive preventive efforts [4,6]. Other regions with high melanoma incidences and ongoing rising trends are Western and Northern Europe, as well as North America [2,4,5]. The increasing incidence rates over recent decades, a better understanding of genetic and environmental risk factors, and a growing general interest in cancer risk prediction have fueled the development of risk prediction models for melanoma. Risk prediction models enable the proper identification of individuals at high risk of developing the disease. They are essential tools for more effective, targeted screenings of individuals at higher risk as a part of secondary prevention strategies.

Although a variety of prediction models for assessing the individual melanoma risk were published over the past 40 years, none have become widely accepted in clinical practice. An essential prerequisite for a reliable risk prediction model that can be implemented in clinical practice, is a properly conducted, well-reported and validated development study. Currently, many risk prediction models are not externally validated [7,8,9], which means that the performance of the model has not been evaluated by an independent dataset. This is important, because shortcomings in study design, methods, conduct, or analysis often lead to overoptimistic predictive performance estimates of the model in the development study [10]. This overoptimism, i.e., the overestimation of the model’s predictive ability, results typically from an overfitting of the developed model to specific characteristics of the dataset that was used to develop the model. When the prediction model is applied to new data, the predictive performance is worse than before [11,12]. This in turn can result in inaccurate models leading to false predictions, which would be detrimental when using the model in clinical practice for risk stratification. False predictions may lead to either unnecessary or insufficient interventions that may influence the health of those affected by the wrong prediction. Thus, it is necessary to evaluate the presence of systematic error in risk prediction studies which may jeopardize the validity of conclusions drawn from such studies. Regarding the assessment of bias in melanoma risk prediction, there is still a need to catch up with other areas of prediction modeling. None of the existing systematic reviews on melanoma prediction studies included a risk of bias (ROB) assessment, which motivated us to fill this gap using the recently developed PROBAST (Prediction model Risk Of Bias ASsessment Tool; https://www.probast.org (accessed on 28 April 2022)) methodology [13].

PROBAST was developed in 2019 to facilitate the tailored ROB assessment for studies exploring prediction models. It provides a methodological quality assessment of primary studies that report on the development, validation, or update of prediction models. The tool can be used for all clinical domains, predictors, outcomes, and modeling techniques [13,14].

The primary objective of this work was to assess the validity of published prediction models for incident cutaneous melanoma using a standardized procedure based on PROBAST and to evaluate the evolution of these assessments over time. In addition to describing the PROBAST results for the overall and domain-specific ratings, we discuss the implications of our assessment results for the current state of knowledge on predicting incident cutaneous melanoma.

## 2. Materials and Methods

### 2.1. Study Selection and Eligibility Criteria

Details on the study selection and eligibility criteria were published previously in a report describing the reporting quality of melanoma prediction studies [15]. In brief, we included studies reporting the development and validation of models for predicting the individual risk of occurrence of cutaneous melanoma. Publications focusing on solely validating and/or updating previously published prediction models were not included. Only studies providing either absolute risks or risk scores, or report mutually adjusted relative risks for primary cutaneous melanoma were eligible. The set of studies to be assessed was based on a recent systematic review of melanoma prediction modeling [7] that updated two earlier systematic reviews on this topic [8,9]. To ensure that our PROBAST rating included all relevant studies, we performed a literature update for the time interval since the end of the search period for the systematic review [7], i.e., February 2020 and August 2021. In particular, the forward snowballing technique [16] was applied to all three systematic reviews [7,8,9] and an electronic literature search in PubMed using the same search string as in [7] was conducted.

### 2.2. PROBAST Rating

The ROB of each study was assessed independently by six reviewers (I.K., S.M., M.V.H., T.S., K.D., O.G.). The reviewer panel was multidisciplinary and consisted of reviewers with methodological (I.K., O.G.), clinical (S.M., M.V.H.), and public health (T.S., K.D.) backgrounds at different levels of experience. All reviewers used the PROBAST tool provided on https://www.probast.org/ (accessed on 28 April 2022). Furthermore, a web-based input tool was created for data collection using the software SoSci Survey version 3.2.21 (SoSci Survey GmbH, Munich, Germany) [17]. All six reviewers assessed all 42 studies. Disagreements between the reviewers regarding the ROB rating were resolved in 10 virtual consensus meetings. In the case of sustained disagreements, two independent referees (A.B.P., W.U.) decided.

The PROBAST tool consists of the four domains: “participants”, “predictors”, “outcome”, and “analysis”. For each domain, the ROB was rated individually as either low, high, or unclear. Several signaling questions that were answered as yes, probably yes, no, probably no, or no information, assisted the reviewer in judging the ROB for each domain. Finally, an overall ROB was assigned to the study based on the ratings in the four domains. According to the given rules in the PROBAST tool [13], the overall ROB is obtained by taking the lowest rating of any domain-specific ROB (“worst score counts principle”). Consequently, the overall ROB was high if at least one of the four domains was rated as high. If at least one domain was judged as unclear and all other domains as low, the overall ROB was unclear. Thus, a study only received a low overall ROB if all four domains were judged as having low ROB. However, according to PROBAST guidance, downgrading to high or unclear ROB should be considered if a prediction model was developed without any external validation. In the absence of external validation, the model evaluation was only considered to be low ROB, if the development was based on a very large dataset and included some form of internal validation.

Since the ROB rating strongly depends on the reviewer’s judgment, some decision rules for the specific setting of melanoma prediction studies were defined by the reviewers in advance to establish a common standard for the rating (see Section 2.3). The decision rules overruled individual ratings and referee decisions. Therefore, all ratings were checked for consistency with the self-defined decision rules and discussed in the case of disagreement.

### 2.3. Description of Domains and Decision Rules

#### 2.3.1. Domain 1: Participants

This domain was related to possible sources of bias associated with the data sources and participant selection. In general, the selection of participants should represent the target population [14]. We defined the following specific rules for this domain: A study received a high ROB if (1) in case-control studies, the cases were recruited in a single center or the controls consisted of hospital controls, (2) in cohort studies, no population sample was used or the study population was self-selected, or (3) in studies based on risk estimates from meta-analyses, the studies included in the meta-analyses met the criteria for a high ROB in this domain. If the references of the studies included in the meta-analyses were not given, the ROB is rated as unclear.

#### 2.3.2. Domain 2: Predictors

The domain “predictors” covered possible sources of bias related to the selection and assessment of predictors. The risk factors had to be defined and collected in the same way for all study participants [14]. Our specific decision rules included that pooled studies and meta-analyses were rated with a high ROB as default, as heterogeneity in definition and assessment of predictors between the included studies was assumed. If it was explicitly described that no heterogeneity existed, e.g., when using identical protocols for the risk factor assessment, a low ROB rating was possible. Furthermore, the use of risk factors with possible recall bias in case-control studies led to an unclear ROB rating. These included predictors related to natural (solar) and artificial UV exposure in the past.

#### 2.3.3. Domain 3: Outcome

The third domain covered a possible bias generated by the definition or determination of the outcome. Objective outcomes, such as histologically confirmed diagnoses, are less susceptible to bias than outcomes that require subjective interpretation or are based on participants’ self-assessment [14]. Consequently, we specified the following rule for ROB ratings: outcomes without verified melanoma diagnosis, e.g., self-reported lifetime melanomas that were assessed via questionnaire, are rated as high ROB.

#### 2.3.4. Domain 4: Analysis

The focus of the last domain was a potential bias in the estimated predictive performance triggered by inappropriate analysis methods or omission of important statistical considerations. Aspects of the analysis to be considered for the bias rating included: (1) whether the sample size was sufficient, (2) whether predictors were incorporated appropriately into the model, (3) whether missing data were handled adequately, (4) whether the predictive performance of the model was evaluated systematically and (5) whether model overfitting was accounted for [14]. We defined the lack of internal and external validation as a sufficient criterion for a high ROB. Another criterion for a high ROB rating was the lack of quantitative information about performance measures. Thus, at least one performance measure and one form of validation had to be reported to obtain a low ROB, provided that the analysis regarding the other aspects was sound. If the analysis contained components whose effect on the results was unclear or the description allowed no definite categorization as either low or high ROB, the domain received an unclear ROB rating. 

#### 2.3.5. General Decision Rules

For all domains, if the information on domain-specific aspects relevant for ROB assessment given in the study publications was too limited to clearly assess the ROB, the respective domain was rated as unclear. Furthermore, an unclear rating was assigned if specific aspects of the study design or methods may lead to bias in the results, but this could not be assessed with certainty based on the information provided by the study publication. The full list of specific decision rules for high and unclear ROB that was updated after the rating and consensus meetings can be found in the Appendix A.

### 2.4. Statistical Analysis

The results of the ROB assessment were analyzed descriptively and presented as absolute and relative frequencies. A possible temporal effect regarding changes in overall and domain-specific ROB rating distributions was additionally investigated. To this end, the studies were divided into three groups based on their year of publication. Using the tertiles of the distribution of publication years we defined the following three time intervals: “1988–2006” (*n* = 14), “2007–2014” (*n* = 15), and “2015–2021” (*n* = 13). We used the Mantel test [18] to check for an association between ROB ratings and time interval as the Mantel test incorporated the ordinal structure of both variables which the Chi-squared test, the statistical standard test in this situation, would have ignored. Due to the sparse data situation we faced in our study, we employed the exact version of the Mantel test based on the network algorithm developed by Mehta and Patel [19]. In addition, we evaluated the presence of a temporal trend in overall and domain-specific ratings also in multinomial logistic regression models. Such an approach avoids the necessity of categorizing publication years into arbitrary intervals as it uses the temporal information in its continuous form as predictor. *p*-values were obtained from likelihood ratio tests to assess the impact of the predictor “publication year”. *p*-values smaller than 0.05 were interpreted as indicating statistical significance. All statistical analyses were performed using the R software version 4.1.1 (R Foundation for Statistical Computing, Vienna, Austria) [20]. Multinomial logistic regression modeling was implemented using the “nnet” package of R [21].

## 3. Results

### 3.1. Study Characteristics

In total, we included 42 studies in our PROBAST rating. Forty studies [22,23,24,25,26,27,28,29,30,31,32,33,34,35,36,37,38,39,40,41,42,43,44,45,46,47,48,49,50,51,52,53,54,55,56,57,58,59,60,61] were adopted from the most recent systematic review about risk prediction models for melanoma that was published in 2020. The remaining two recent studies [62,63] were identified through the updated literature search. Study characteristics are summarized in Table A1 in the Appendix B. Thirty-five of the 42 studies (83%) solely described the development of a melanoma risk prediction model, while seven studies (17%) reported both development and external validation. The publication years of the studies ranged from 1988 to 2021, with a pronounced increase in the number of studies in the last decade of this interval. The majority of studies were case-control studies (*n* = 30). Ten studies used a cohort study design and two studies used published material from meta-analyses to determine predictors and risk estimates.

### 3.2. Results of Risk of Bias Rating

Results of the domain-specific and overall ROB ratings of our set of 42 studies are shown in Figure 1. The individual ROB ratings of all studies are included in Table A1. In the following, the results for the individual domains are described.

#### 3.2.1. Domain 1: Participants

In the “participants” domain, 24 studies (57%) were rated as high, three studies (7%) as unclear, and 15 studies (36%) as low ROB (see Figure 1). In 15 studies, the selection of controls in case-control study designs was decisive for the high ROB rating, mainly because of the use of hospital controls (*n* = 14). In addition, four studies based on meta-analyses received a high ROB as they each contained studies with a high ROB. In four cohort studies, the use of a self-selected screening population resulted in a high ROB rating. Further reasons that led to an unclear or high ROB rating are listed in Table 1.

#### 3.2.2. Domain 2: Predictors

Three studies (7%) were rated as high ROB in the “predictors” domain due to heterogenous predictor assessment of studies included in the meta-analyses or pooled studies (Figure 1, Table 2). Furthermore, 27 studies (64%) were rated as unclear. In the majority of cases (*n* = 21) the reason was potential recall bias in case-control studies due to predictors related to UV exposure in the past. Three studies did not provide enough information for the evaluation of potential bias which also lead to an unclear ROB rating. The remaining three studies with an unclear ROB rating in the predictors domain suffered from discrepancies between development and validation datasets. Twelve (29%) of the included studies were rated as low ROB.

#### 3.2.3. Domain 3: Outcome

The “outcome” domain comprised the highest proportion (*n* = 37, 88%) of low ROB ratings among all four domains in our investigation. The ROB of one study (2%) was rated as unclear due to limited information regarding the definition and assessment of the outcome (Figure 1, Table 3). Four studies (10%) received a high ROB rating. Three of the four studies did not use verified outcomes: self-reported lifetime melanomas (*n* = 2) or suspected melanomas (*n* = 1). The fourth study used a composite outcome consisting of melanoma and cannot-exclude-melanoma/severely dysplastic nevi.

#### 3.2.4. Domain 4: Analysis

In the “analysis” domain, eight studies (19%) had an unclear ROB, whereas for 20 studies (48%) the ROB was rated as high and for 14 studies (33%) as low (Figure 1). Reasons for an unclear ROB rating were, e.g., limited information regarding the analysis (*n* = 4) and non-standard handling of predictors during the statistical analysis entailing unknown impact on the results (*n* = 2), see Table 4. The main reason for high ROB was a missing internal and external validation (*n* = 19). In several cases, multiple reasons for a single study led to a high ROB rating. However, in Table 4 we only listed the reasons that were decisive for our rating, which was primarily the lack of validation. The lack of internal and external validation often occurred in combination with missing performance measures (*n* = 12), a limited sample size (*n* = 3) and/or missing information regarding one or multiple aspects of the analysis (*n* = 14).

#### 3.2.5. Overall ROB

Overall, only one study (2%) received a low ROB rating, whereas seven studies (17%) were judged to have an unclear ROB. Four [27,28,58,59] of these seven studies received their unclear ROB rating due to an unclear ROB rating in a single domain, while the remaining three studies [31,51,61] had an unclear ROB rating in two domains. The majority of studies (*n* = 34; 81%) were associated with a high ROB (Figure 1). For one study [55], we used the option of downgrading according to PROBAST guidance. The study received a low ROB rating in the domains “participants”, “outcome” and “analysis”, and an unclear rating in the “predictors” domain that would have resulted in an overall unclear ROB accordingly. However, due to its small sample size and lacking external validation the study was downgraded to high ROB.

### 3.3. Temporal Analysis

The proportion of studies with low, unclear, and high ROB ratings in the three time intervals is visualized in Figure 2. A more detailed visualization of the distribution of all ROB ratings over time can be found in the Appendix A. For the domain “analysis” we found a clear temporal trend toward better ROB ratings for more recent studies. The proportion of studies rated as high ROB decreased significantly over the three time intervals (79% vs. 40% vs. 23%, *p* = 0.001). This finding was corroborated by the results from multinomial logistic regression modeling identifying the publication year as a significant predictor (*p* = 0.004). For the three other domains we did not observe such a clear-cut temporal development of ROB rating distributions and the statistical analyses did not point to a significant effect of publication year on these domain-specific ROB ratings. The overall ROB rating distribution indicated some improvement over time: the proportion of studies rated as high ROB decreased steadily from 93% in 1988–2006 over 80% in 2007–2014 to 69% in 2015–2021, but this decline missed statistical significance in the categorical analysis and the multinomial logistic regression analysis.

## 4. Discussion

The results of our ROB assessment showed a clear deficit of valid risk models for melanoma prediction, as the vast majority (81%) of the included 42 studies was associated with a high ROB. Thus, the evidence base of high-quality studies that can be used to draw conclusions on the prediction of incident cutaneous melanoma is currently much weaker than the high number of studies on this topic would suggest.

Only one [50] of the 42 studies had a low overall ROB score. The study was the QSkin Sun and Health Study, a prospective cohort study of 43,794 participants randomly sampled from the population of Queensland, Australia in 2011 [64]. Up to now, the QSkin study is the largest prospective study ever conducted specifically to address melanoma and other skin cancer outcomes. The study report from 2018 [50] described separately the prediction of invasive and any melanoma (incl. in situ melanoma) using self-assessed risk factors. The model for predicting invasive melanoma included the following seven risk factors: age, sex, tanning ability, number of nevi at 21 years of age, hair color, number of actinic skin lesions destroyed, and sunscreen use when outdoors in the past year. The same risk factors were also part of the prediction model for any melanoma that additionally included five risk factors, e.g., family history of melanoma and number of skin checks by a doctor in the past three years. Although the study raised no concerns regarding systematic error in study design, conduct, methods, and analysis, the application of its risk models in clinical practice is limited by their moderate predictive performance: The model discrimination, as described by the C-index, was only 0.69 (95%-CI: 0.62, 0.76) for the invasive melanoma model and 0.72 (95%-CI: 0.69, 0.75) for the any melanoma model showing that additional explanatory variables are required to improve the predictive performance.

Furthermore, four publications [27,28,58,59] had overall an unclear ROB score resulting from a domain-specific unclear ROB rating in a single domain (in all four cases the domain “predictors”). These publications described externally validated models from the same population-based case-control study. In all four publications data from the Australian Melanoma Family Study [65] were used to develop the prediction model. This study only included cases diagnosed with invasive cutaneous melanoma at age 18–39 years and is therefore highly selective. Data from the Leeds Melanoma Case-Control Study [66] were used to validate the model (in [59] data from three additional case-control studies served for additional external validations). Two of the publications incorporated [27,28] genotype information, while the remaining two [58,59] focused on non-genetic risk factors. The difference between the two non-genetic prediction models related to the inclusion of only self-reported risk factors in [59] and the use of physician-assessed risk factors related to skin phenotype in [58]. The models differed considerably in their performance, the AUC describing model discrimination ranged from 0.66 (95%-CI: 0.63, 0.68) for the model including only self-assessed risk factors without genotype information [59] to 0.79 (95%-CI: 0.76, 0.81) for the model including physician-assessed risk factors and genotype information related to the MC1R genotype [28]. The main driver of the increments in the AUC was the incorporation of physician-assessed nevi counts instead of self-assessed nevi density. The use of genotype information had only a moderate impact, contrary to what one would expect from the increasing popularity of genetic risk factors in recent years.

The selection of risk factors has not only a significant impact on the performance of the model, but is also related to possible bias, especially in case-control studies. The high proportion of studies with an unclear ROB rating in the “predictors” domain resulted primarily from the use of predictors related to past UV exposure. Whenever such predictors are ascertained in retrospective case-control studies, estimation of their impact on melanoma risk is prone to recall bias, i.e., a special form of exposure misclassification in case-control studies. For melanoma, the presence of recall bias attracted considerable attention and has been analyzed using different approaches in various studies [67,68,69,70,71,72,73]. There has been no clear conclusion regarding the magnitude of the bias [74,75]. The consequences of incorporating such predictors into melanoma prediction models have not been discussed by any of the developers of these models and remain unclear. Another source of bias in case-control studies that led to most high ROB ratings in the “participants” domain is the use of hospital controls. To prevent bias in case-control studies, the controls must be selected independent of exposure and need to represent the study population at risk of becoming cases [76]. Although the selection of hospital controls has some practical advantages, e.g., they are readily accessible and usually cooperative, the presence of unsuspected associations between the reason for hospital visit and the factors of interest can lead to systematically distorted estimates [77,78,79]. Hospital controls are likely to have a higher frequency of hazardous exposures compared to the general population [80].

The large numbers of high and unclear ROB ratings demonstrate the need to reduce bias in future studies. One possibility is to consider the criteria of ROB tools already in the study planning stage. Thus, sources of bias related to the selection of the study population and the definition of outcome assessment, for example, could be avoided. Another opportunity for reducing bias can be found in the “analysis” domain. The main reason for high ROB ratings was the lack of validation (internal or external), often combined with missing evaluation of model performance. However, we saw a positive temporal trend in this domain: The proportion of high ROB ratings has significantly decreased by more than 50%. This development shows that the journals have been more rigorous in applying pertinent quality standards in recent years, particularly concerning the methodology employed during statistical analysis. An important additional contribution to the positive development is made by the large number of checklists and accompanying guidance papers that have been published in recent years. These include reporting guidelines such as TRIPOD (Transparent Reporting of a multivariate prediction model for Individual Prognosis Or Diagnosis) [81], which provides a checklist of 22 items essential for transparent reporting of a prediction model study [15]. It ensures that all relevant key details on the development process and model performance, which are needed to objectively appraise the validity and usefulness of the model, are reported. Furthermore, guidelines that directly include ROB tools, such as the PRISMA (Preferred Reporting Items for Systematic Reviews and Meta-Analyses) checklist [82] for systematic reviews and meta-analyses, strengthen the focus on the ROB of studies. PRISMA is already required by many scientific journals, which has demonstrably improved the conduct and reporting of systematic reviews and meta-analyses [83]. Other tools for the assessment of ROB are, e.g., the Cochrane ROB tool [84] for randomized controlled trials, which was published in 2011 and updated in 2019 [85]. All of these have the potential to ensure a high transparent quality of studies developing risk prediction models if applied properly. However, we conclude from our results that in order to better implement and advance knowledge about melanoma risk prediction, it is essential to expand the application of existing guidelines in practice to improve the quality of prediction model studies, especially regarding study design and standardization of methodology to conduct this type of studies.

To the best of our knowledge, this is the first assessment of bias in melanoma prediction studies, hence there is no direct comparison of our results with other papers. However, comparisons with ROB results from assessments in other clinical domains are possible. The two systematic reviews by Sassano et al. [86] and Su et al. [87] addressing risk prediction of colorectal cancer and caries, respectively, involved ROB assessment with PROBAST. Both criticized an insufficient number of high-quality studies in their clinical domains, the proportion of studies with high ROB being 94% and 78%, respectively. In 2021 a meta-review by de Jong et al. [88] including 50 systematic reviews across various clinical domains that all used PROBAST for ROB assessment was published. The ROB rating from a total of 1510 individual studies was reported. Similar to our results, the authors observed predominantly unclear and high ROB ratings at the domain-specific levels, while results of the overall ROB were not reported. The domain “analysis” showed with 69% the highest proportion of high ROBs, which is higher than in our rating where the proportion of high ROBs in this domain was 48%. Unlike ours, the results were stable over time. This shows that the positive temporal trend toward higher quality standards concerning statistical methodology, which is visible in melanoma prediction studies, has not yet reached all clinical domains.

During our assessment, we encountered some obstacles in the practical application of PROBAST, which show that the tool is not easily applicable in all situations. According to PROBAST, case-control studies do not represent appropriate data sources and should be rated with high ROB as default. Though case-control studies are more prone to bias, this is not primarily due to the study design itself but due to practical problems in study conduct, some of which have already been described above. Per se, case-control studies can yield results as valid as cohort studies, if they are properly planned, conducted, and analyzed [89]. In addition, some signaling questions that should support the ROB rating, such as the questions “Was the outcome determined without knowledge of predictor information?” and “Was the time interval between predictor assessment and outcome determination appropriate?” in the “outcome” domain, are only applicable for prospective studies. In case-control studies the outcome status is already known when the participants are being selected and thus before the predictor assessment. In general, the continuous adaptation and improvement of rating tools is necessary to further increase their applicability and popularity. In particular, the PROBAST tool should therefore be amended or supplemented for study design-specific features to ensure unequivocal assessment. Otherwise, systematic reviews employing PROBAST need to redefine generic signaling questions for their application.

Due to the above-mentioned obstacles in the applicability of the tool to case-control studies, which accounted for 71% of our included studies, but also to provide a consistent basis for our rating, we defined some specific decision rules that overruled the decisions of individual raters and those of the referees. Since the decision rules were designed to the best of our knowledge but were not validated separately, this may have resulted in some bias in our ROB ratings and constitutes a limitation of our work. Additionally, the ROB judgment is subjective and does not lend itself to a clear objective rating. As different raters may have come to different conclusions on how to rate the individual PROBAST domains, it cannot be ruled out that another group of raters would have come to other results regarding the PROBAST ratings in the same set of melanoma prediction studies. We tried to minimize this rater dependence by defining the decision rules, by holding consensus meetings to resolve discrepancies in ratings, and by involving two independent referees in the case of persisting disagreement. Another limitation is that the studies assessed in our rating do not cover all studies dealing with melanoma prediction. The basis for our set of studies were three systematic reviews that we supplemented with a literature update. Nevertheless, due to the eligibility criteria of the systematic reviews, we included in our assessment only studies reporting (i) solely the development and (ii) both the development and external validation of a melanoma risk prediction model. Thus, studies focusing exclusively on external validation or update of preexisting models, for which PROBAST is also designed, were not part of our investigation. The segment of such studies is, however, not strongly represented in melanoma research. We are aware of only three studies [90,91,92] that exclusively addressed external validation of previously published models and none that updated a published model. Results of our investigation do not allow conclusions regarding ROB in these study types.

## 5. Conclusions

In conclusion, the vast majority of studies on melanoma risk prediction models had a high ROB rating showing that the validity of published prediction models for incident cutaneous melanoma was poor. The selection of participants and the omission of appropriate validation efforts in the statistical analyses were frequent sources of bias. A low ROB is a necessary prerequisite for any prediction model to be used reliably in practice. As a consequence, there is currently only a thin evidence base of high-quality studies to predict melanoma risk yet. However, some positive temporal trend in bias reduction inspires hope that this may change in the future.

## Figures and Tables

**Figure 1 cancers-14-03033-f001:**
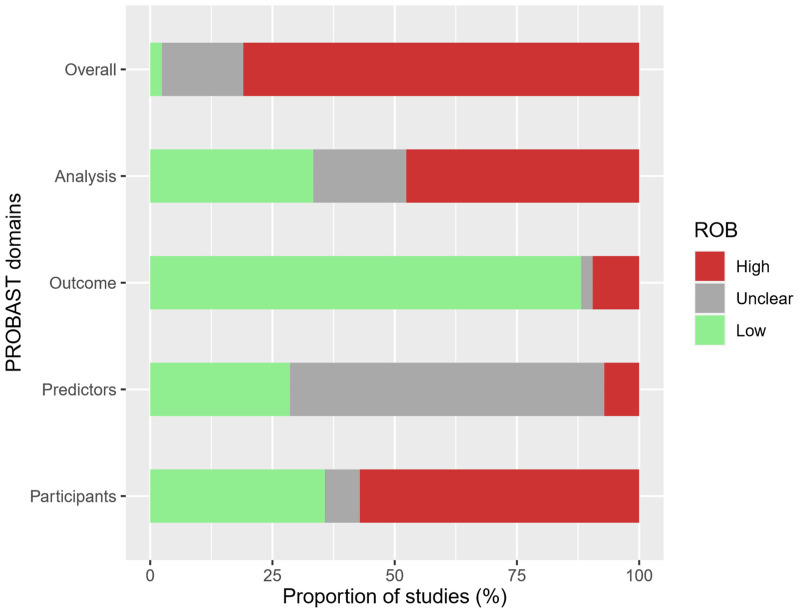
Risk of bias rating overall and per domain (*n* = 42 studies).

**Figure 2 cancers-14-03033-f002:**
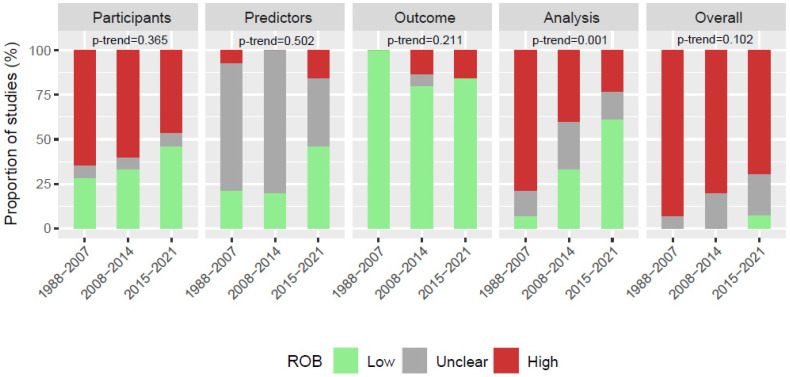
Comparison of proportions of studies with high, unclear, and low ROB for domain-specific and overall ratings in the three time intervals “1988–2006” (*n* = 14),”2007–2014” (*n* = 15) and ”2015–2021” (*n* = 13). *p*-values for trend were obtained using the exact Mantel test.

**Table 1 cancers-14-03033-t001:** Reasons for unclear (*n* = 3) and high (*n* = 24) ROB ratings in the “participants” domain.

Unclear ROB		High ROB	
Reason	*n* (%)	Reason	*n* (%)
Limited information	2 (67%)	Hospital controls (case-control studies)	14 (58%)
Data from a customer data-base offering genetic analyses without information regarding population coverage	1 (33%)	Meta-analysis including studies with high ROB	4 (17%)
		Self-selected screening population/no population sample (cohorts)	4 (17%)
		Highly selected sample	1 (4%)
		Mixed bag of controls (including hospital controls)	1 (4%)

**Table 2 cancers-14-03033-t002:** Reasons for unclear (*n* = 27) and high (*n* = 3) ROB ratings in the “predictors” domain.

Unclear ROB		High ROB	
Reason	*n* (%)	Reason	*n* (%)
Potential recall bias	21 (78%)	Pooled study or meta-analysis with heterogenous predictor assessment	3 (100%)
Limited information	3 (11%)		
Replacement of predictors in validation	1 (4%)		
Unclear harmonization of predictor variables in development and validation datasets	1 (4%)		
Missing predictors in validation dataset	1 (4%)		

**Table 3 cancers-14-03033-t003:** Reasons for unclear (*n* = 1) and high (*n* = 4) ROB ratings in the “outcome” domain.

Unclear ROB		High ROB	
Reason	*n* (%)	Reason	*n* (%)
Limited information	1 (100%)	Self-reported outcome	2 (50%)
		Composite outcome (melanoma and severely dysplastic naevus)	1 (25%)
		Suspected melanoma as outcome	1 (25%)

**Table 4 cancers-14-03033-t004:** Reasons for unclear (*n* = 8) and high (*n* = 20) ROB ratings in the “analysis” domain.

Unclear ROB		High ROB	
Reason	*n* (%)	Reason	*n* (%)
Limited information	4 (50%)	No validation	19 (95%)
Non-standard handling of predictors during the analysis	2 (25%)	Limited sample size	1 (5%)
Rounding of model coefficients to define the risk score	1 (12.5%)		
Several aspects of analysis unclear	1 (12.5%)		

## Data Availability

Not applicable.

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
