# Peer review of "Using the Prediction Model Risk of Bias Assessment Tool (PROBAST) to Evaluate Melanoma Prediction Studies"

_cancers, 2022, doi:10.3390/cancers14123033_

Round 1
Reviewer 1 Report
The authors, to assess the validity of published prediction models for incident cutaneous melanoma, use a standardized procedure based on PROBAST (Prediction Model Risk of Bias Assessment Tool). PROBAST assesses the risk of bias regarding the applicability of primary studies that developed or validated multivariate prediction models for diagnosis or prognosis.
The article is well written, the methodology is correct and the topic is current. Critical reading of melanoma incidence studies is necessary to assess the presence of systematic error in risk prediction studies and assessment of biases in melanoma risk prediction that may jeopardize the validity of conclusions drawn from these studies.
However, there are some aspects that could be improved or clarified:
Only the graphs appear in the figures and there should be a more detailed legend that should include the statistical comparison test and the p-value
For each domain, the ROB was rated individually as either low, high, or unclear. The differences between high and low are clear, but not between low and unclear. For example: Unclear harmonization of predictor variables in development and validation datasets, missing predictors in validation dataset, data from a costumer data-base offering genetic analyzes without information regarding population coverage. Why are they not considered low ROB and have been considered unclear?
Reviewer 2 Report
The manuscript judges the risk of bias of 42 melanoma prediction studies and addresses a gap in the literature. However, the exclusion of studies focusing exclusively on external validation or updates of preexisting models limits the informative value of the paper considerably.
- I only realized in the last sentence before the conclusion that studies focusing exclusively on external validation or updates of preexisting models are existing, but not included in this work. First, please make this clear to reader much earlier. Second, this exclusion limits the informative value of the paper considerably. Without information regarding further external validations and improvements of predictions models, it is not possible to make a general statements of ‘poor validity of published prediction models for incident cutaneous melanoma’.
- The results section contains results for statistical tests on (1) changes over time and (2) comparisons between only-development-studies and development-and-validation-studies. Statistical tests should only be applied for hypotheses that are announced as research questions beforehand.
- Does it make sense to conduct hypothesis tests for the ‘analysis’ domain for the null hypothesis ‘there is no difference in ROB of only-development-studies and development-and-validation-studies’? As I can see from the supplemental table, studies without validation are automatically categorized as ‘high ROB’. That means that H0 is essentially: In truth, all development-and-validation-studies have a high ROB in the ‘analysis’ domain. I think we can agree that we can reject that hypothesis without statistical testing.
The case is similar for ‘overall ROB’. - For analysis of changes over time, calendar years are categorized into three groups. Categorization of continuous variables should be avoided whenever possible. In the situation here, the categorization is not persuading because it is not based on some reasonable argument (such as before/during/after some intervention), it is only motivated through the number of included studies.
Could you fit a regression for the categorical outcome ROB (high/low/unclear) and the continuous predictor ‘time’ instead, for example with a multinomial logistic regression? - It says in the conclusion that ‘The resulting thin evidence base of high-quality studies made it impossible to answer the question of how to predict melanoma yet’. This sounds as if the decision for a prediction model depends only on a low ROB, when in fact several other factors play a role, such as the available data (self-report only, doctor's report, laboratory data or a combination of these), the population of interest (skin color, country, age group…) and desired outcome (score, sex-specific score, score not including age, …).
- For better readability, please use quotation marks when referring to the name of domains (e.g. the domain ‘analysis’).
Round 2
Reviewer 2 Report
The authors have considered all comments thoroughly, explained their decisions and revised the manuscript accordingly. I have no further remarks.